# SketchFill: Sketch-Guided Code Generation for Imputing Derived Missing Values

## Abstract

Missing value is a critical issue in data science, significantly impacting the reliability of analyses and predictions. Missing value imputation (MVI) is a longstanding problem because it highly relies on domain knowledge. Large language models (LLMs) have emerged as a promising tool for data cleaning, including MVI for tabular data, offering advanced capabilities for understanding and generating content. However, despite their promise, existing LLM techniques such as in-context learning and Chain-of-Thought (CoT) often fall short in guiding LLMs to perform complex reasoning for MVI, particularly when imputing **derived missing values**, which require mathematical formulas by considering data values across rows and columns. This gap underscores the need for further advancements in LLM methodologies to enhance their reasoning capabilities for derived missing values. To fill this gap, we propose **SketchFill**, a novel sketch-based method to guide LLMs in generating accurate formulas to impute missing numerical values. SketchFill first utilizes a general user-provided **Meta-Sketch** to generate a **Domain-Sketch** tailored to the context of the input dirty table. Subsequently, it fills this Domain-Sketch with formulas and outputs *Python code*, effectively bridging the gap between high-level abstractions and executable solutions. Additionally, SketchFill incorporates a **Reflector** component to verify the generated code. This Reflector assesses the accuracy and appropriateness of the outputs and iteratively refines the Domain-Sketch, ensuring that the imputation aligns closely with the underlying data patterns and relationships. Our experimental results demonstrate that SketchFill significantly outperforms state-of-the-art methods, achieving **56.2%** higher accuracy than CoT-based methods and **78.8%** higher accuracy than MetaGPT. This sets a new standard for automated data cleaning and advances the field of MVI for numerical values.

## 1 Introduction

Missing value imputation (MVI) represents a longstanding data quality challenge, critically impacting the reliability and effectiveness of data-driven industries, such as healthcare (Shetty et al., 2024; Psychogyios et al., 2023), IoT research (Adhikari et al., 2022; Li et al., 2023c), and spatial time-series analysis (Wu et al., 2015; Gong et al., 2023; Tashiro et al., 2021). A notable aspect of this challenge is the substantial amount of time and resources it demands (Rezig et al., 2019; Rashid & Gupta, 2020). The State of Data Science 2020 Survey, made by Anaconda[1], revealed that on average 45% of time is spent getting data ready (19% and 26% for loading and cleaning respectively) before the data scientists can use it to develop models and visualizations. This not only consumes an excessive amount of human resources but also significantly slows down the analytical processes in data-centric corporations.

Given its critical importance, MVI has been extensively explored within the academic and professional communities. The literature is rich with a variety of approaches, ranging from traditional statistical methods to more contemporary machine learning and deep learning-based techniques. Despite the advancements, the task of MVI continues to pose significant challenges, largely due to the need for substantial domain-specific expertise to accurately handle missing data. In this work, we address a specific subset of this problem, termed **Derived Missing Value Imputation (DMVI)**, which can

---

[1]https://www.anaconda.com/resources/whitepapers/state-of-data-science-2020

| | BMI (Simple) | | | Supermarket (Intermediate) | | | | Bajaj Finance (Complex) | | |
|---|---|---|---|---|---|---|---|---|---|---|
| | Height | Weight | BMI | Unit Price | Quantity | Tax5 | Total | Period | Close | SMA5 |
| | 1.85 | 109.3 | 31.936 | 16.67 | 7 | 5.83 | 122.52 | 09:50 | 408.78 | 409.27 |
| | 1.58 | 97.1 | 38.896 | 73.96 | 1 | 3.7 | 77.66 | … | … | … |
| | 1.71 | 79.32 | 27.126 | 28.32 | 7 | 7.08 | 146.68 | 10:30 | 409.19 | 409.142 |
| | 1.73 | 74.12 | NaN | 30.68 | 3 | 4.6 | NaN | 10:40 | 410.03 | NaN |

$$BMI = \frac{Weight}{Height^2} \qquad Total = UnitPrice \cdot Quantity + Tax5 \qquad SMA5 = \frac{1}{5}\sum_{i=0}^{4} Close_{t-i}$$

Figure 1: DMVI samples from our experimental datasets. The `NaN` represents the missing values and the formula on the bottom is the derived solution for the missing value imputation.

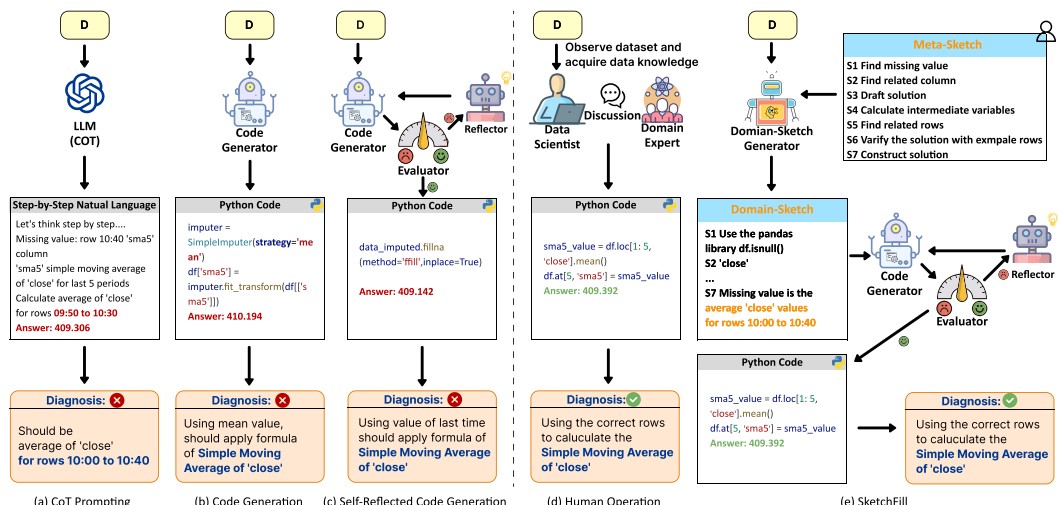

Figure 2: Different LLM-based approaches for DMVI

often be observed in real-world numerical data as shown in Figure 1. The derivation process is often tightly coupled with the characteristics of both the domain and the dataset, implying that imputation methods effective in one context may not generalize to others. Consequently, both domain-specific knowledge and dataset-specific knowledge are essential to carry out DMVI tasks.

In recent years, large language models (LLMs) have shown considerable promise in addressing complex data processing tasks (Zhao et al., 2023; Zhang et al., 2023a), particularly in the field of table understanding in knowledge extraction and content generation (Sui et al., 2024; Zha et al., 2023; Li et al., 2023b). Their ability to understand and manipulate textual and, increasingly, tabular data suggests a promising avenue for enhancing MVI techniques. By leveraging the extensive knowledge embedded within LLMs (Razniewski et al., 2021), there is potential to significantly improve the accuracy and efficiency of the DMVI task, where understanding nuanced data relationships and contexts is crucial.

Let's illustrate the limitations of existing methods using an example. Figure 2 illustrates various approaches that impute missing values (annotated with `NaN`) in the dirty data $D$ using LLMs. Note that **SMA5** is a formula commonly used in the financial domain that requires aggregating the previous five rows (See the *Bajaj Finance* sample in Figure 1).

*Baseline-1: Chain-of-thoughts (CoT) for DMVI.* Figure 2(a) explores the use of CoT prompting, which encourages LLMs to process information step-by-step. This method, however, tends to produce answers that are "reasonable" yet oftentimes not sufficiently accurate.

*Baseline-2: Code Generation for DMVI.* Figure 2(b) highlights the code generation capabilities of LLMs, which tend to employ common functions such as calculating the *mean*. Though straightforward, this approach often fails to address the complexities inherent in many DMVI scenarios.

*Baseline-3: Self-Reflected Code Generation for DMVI.* Figure 2(c) introduces the use of a Reflector to refine the generated code. Despite this enhancement, it still struggles to support LLMs in making intricate reasoning required for formula generation.

**Rethink: How Do Data Scientists Perform DMVI?** As illustrated in Figure 2(d), the data scientist often starts his/her observation on the missing value and surrounding rows and columns (Gibson et al., 1989) to build dataset knowledge essential for DMVI. Witten & Frank (2005) also mention the importance of collaboration with domain experts to build domain knowledge. These expertises are then translated into algorithms via coding, enabling the automation of the DMVI process and ensuring accurate results. Inspired by this human operation, we argue that the explicit manifestation of both dataset and domain knowledge via a Meta-Sketch can guide LLMs to generate accurate and tailored Python code for DMVI.

**Our Proposal: Sketch-Guided Self-Reflected Code Generation.** Figure 2(e) presents our proposed method SketchFill. Accordingly, SketchFill mimics this human-driven expertise, which shifts from CoT to using an explicit user-provided Meta-Sketch to direct LLMs in generating a Domain-Sketch. Consequently, the Domain-Sketch, which contains the reasoning result embedded with knowledge of the particular dataset, can better guide LLMs to generate Python code for DMVI by Code Generator. It also adopts a Reflector module to iteratively refine the Domain-Sketch, leading to the accurate formulation of a problem-specific formula that is subsequently instantiated into Python code. Afterwards, the Summarizer module will wrap the imputation code into a structured format for execution. This approach significantly improves the precision and applicability of DMVI solutions.

**Contributions.** Our main contributions in tackling DMVI issue are summarized as follows:

1. **Integration of Meta-Sketch and Domain-Sketch for DMVI Code Generation:** SketchFill incorporates a high-level user hint Meta-Sketch (e.g., explicitly saying that more rows and more columns need to be checked), which will be used by the Domain-Sketch Generator to produce a Domain-Sketch and then by Code Generator to generate executable Python code. This two-step sketch generation and guided code generation can not only aid LLMs in identifying the correct formulas for imputation but also impose constraints on the output format, ensuring that the results are structured and predictable.

2. **Iterative Reflector-Based Framework:** SketchFill employs an effective reflector-based iterative framework that guides LLMs in discovering the correct formula for imputing missing values. This framework iteratively refines the output, enabling the LLM to align more closely with the complexities of the specific imputation task.

3. **Output Summarization:** To enhance the usability of the imputed data, SketchFill includes a Summarizer that processes the outputs for multiple missing values. Thanks to the constrained output format provided by the Meta-Sketch, the Summarizer efficiently organizes the results into an easily readable format, allowing users to understand and apply the formulas across multiple instances of missing data.

4. **Empirical Validation:** We have conducted comprehensive experiments across five different domains to validate the effectiveness of SketchFill. These experiments demonstrate the robustness and versatility of our approach in improving the accuracy and reliability of missing value imputation across diverse datasets and application contexts.

## 2 RELATED WORK

Missing Value Imputation (MVI) has been explored with the advent of data science, aiming to look at the patterns and mechanisms that create the missing data, as well as a taxonomy of missing data in datasets (Little & Rubin, 2019). Despite a single imputation method like Hot-Deck (Andridge & Little, 2010; Christopher et al., 2019), which is imputed from a randomly selected similar record, we categorize existing solutions in the context of statistics and machine intelligence for missing value imputation as follows.

**Statistic-based Methods.** Initially, the most common strategy for MVI is using a descriptive statistic, e.g., mean, median, or most frequent, along each column, or using a constant value. It is widely

adopted in existing packages and tools, such as *sklearn.SimpleImputer* (Pedregosa et al., 2011) and *Excel FlashFill* (Gulwani, 2011). Besides, curated packages such as MICE (Van Buuren & Groothuis-Oudshoorn, 2011) can impute incomplete multivariate data by chained equations.

**Machine Learning-based Methods.** Tao et al. (2004) improve algorithms in Reverse kNN, allowing to retrieve an arbitrary number of neighbors in multiple dimensions. Sridevi et al. (2011) propose ARLSimpute, an autoregressive model to predict missing values. Stekhoven & Bühlmann (2012) propose an iterative imputation method MissForest that can impute the missing value of mixed-type data. Tsai et al. (2018) propose CCMVI, which calculates the distances between observed data and the class centers to define the threshold for later imputation. Razavi-Far et al. (2020) propose kEMI and kEMI$^+$ for imputing categorical and numerical missing data correspondingly. They both first utilize the k-nearest neighbors (KNN) algorithm to search the K-top similar records to a record with missing values, then invoke the Expectation-Maximization Imputation (EMI) algorithm, which uses feature correlation among the K-top similar records to impute missing values.

**Deep Learning-based Methods.** Gondara & Wang (2018) propose a multiple imputation model based on overcomplete deep denoising autoencoders, which is capable of handling different missing situations in terms of the data types, patterns, proportions, and distributions. With the advent of the diffusion model, the CSDI (Tashiro et al., 2021) acts as a time series imputation method that utilizes score-based diffusion models to exploit correlations on observed values. Later on, Zheng & Charoenphakdee (2022) explore the use of conditional score-based diffusion models for tabular data (TabCSDI) to impute missing values in tabular datasets. Their study evaluates three techniques for effectively handling categorical variables and numerical variables simultaneously.

**LLM-based Methods.** With the advance of LLM, especially superb generative models like GPT, some techniques can be applied to the MVI task. Since LLMs are trained on extensive and diverse corpora, they inherently possess knowledge of a wide array of common entities (Razniewski et al., 2021; Narayan et al., 2022), intuitively, we can directly ask LLM to perform MVI given some dirty data. Besides, CoT prompting (Wei et al., 2023) can significantly improve the ability of complex reasoning. Alternatively, Poldrack et al. (2023) explore the code generation ability utilizing LLM so that a specific script for MVI can be generated. Additionally, LLM-powered agentic tool, MetaGPT (Hong et al., 2023; 2024) reveals its capability on data cleaning tasks. In short, we will seize the above methods and discuss their performance in the later section.

## 3 METHODOLOGY

### 3.1 PRELIMINARIES

**Derived Missing Value Imputation.** Let $T$ be a table with missing values that are denoted by `NaN`. The problem of *derived missing value imputation* (DMVI) can be mathematically expressed as finding a formula $f$ such that for each missing value `NaN` in $T$, we have:

$$\mathbf{x} = f(T), \ s.t. \ \mathbf{x} \approx \mathbf{x}_g \qquad (1)$$

where $\mathbf{x}$ is the filled value and $\mathbf{x}_g$ is the ground truth value. Moreover, the complexity of the deriving formula involving missing values can range from simple to highly intricate as shown in Figure 1.

**Sketch-Guided Approach.** Normally, the sketch refers to a high-level, abstract representation of the content. The sketch-guided approach has been proven to be effective in various scenarios, such as code generation (Li et al., 2023a; Zan et al., 2022; Calò & Russis, 2022), text-to-SQL (Choi et al., 2020), and image generation (An et al., 2023). In our framework, we adopt it to guide LLMs for better code generation by incorporating two types of sketches: Meta-Sketch and Domain-Sketch, respectively. A **Meta-Sketch** is defined as a series of instructions that mimic the task-specific procedure of expert users. A **Domain-Sketch** is iteratively generated into a series of curated instructions from the Meta-Sketch. We will elaborate them in Domain-Sketch Generator Section 3.3.

**Prior Analysis.** We conducted several observations prior to our framework as shown in Figure 2. Intuitively, LLMs can be leveraged to fill missing values by a simple prompt "fill the missing values of the input dirty table", or using Chain-of-Thought by adding a prefix "let's think step by step".

---

**Algorithm 1:** SketchFill imputation workflow

---

**Input:** Meta-Sketch, $T$ and $V$
**Output:** $D'$ and $F$

1  $C, D = \{T_1, T_2, \cdots, T_n\} \leftarrow$ randomly sample k rows of clean data from $T$ and chunk $T$;
2  $C_m \leftarrow$ randomly masked $\lambda$ rows of of $V$ in $C$;
3  **while** $retry \leqslant retry\_limit$ **do**
4       $S, P \leftarrow$ **Call Domain-Sketch Generator:** use Meta-Sketch to generate Domain-Sketch $S$
       and **Call Code Genrator:** interpret $S$ into Python code $P$ to impute missing value in $C_m$;
5       $C'_m \leftarrow$ Execute $P$ to get imputed data;
6       **if** *Call Evaluator:* $C'_m = C$ **then**
7           $F \leftarrow$ **Call Summarizer:** generate imputation function for $V$ in form of Python code
         function $F$;
8       **else**
9           $S_{new} \leftarrow$ **Call Reflector:** reflect and generate new Domain-Sketch $S_{new}$;
10          $retry = retry + 1$;
11      **end**
12 **end**
13 $D' \leftarrow$ Execute $F$ on $D$;
14 **return** $D'$ and $F$;

---

However, these approaches may not be robust enough to guide LLMs to derive a correct solution for the DMVI task, such as utilizing formulas. Besides, directly applying code generation also fails to generate the correct solution for the DMVI task, as it lacks some high-level user hint to unleash the power of LLMs to reason the domain knowledge of the dataset. We also include the reflector to polish the generated code. However, without the guidance from the sketch, the reflector lacks of DMVI task understanding so that fails to carry out the correct inference and refine the Python code. To cope with this problem, we offer a sketch-guided solution. On the one hand, it allows the users to provide simple hints. On the other hand, it can provide more context information to LLMs, in order to perform targeted code generation and improve the reflection.

### 3.2 SKETCHFILL OVERVIEW

We propose a novel framework that leverages extensive knowledge embedding within LLMs to resolve the challenge of DMVI. Algorithm 1 elaborates the workflow by which we implement SketchFill. It takes the input of a dirty table $T$, a user-provided prompt Meta-Sketch, and the column $V$ that requires for imputation. It outputs an imputed dataset $D'$ and a Python code function $F$ for human review.

To extract the formula within the dirty table, SketchFill first samples a subset of clean data $C$ within the dirty table and randomly masks it (lines 1-2). Then SketchFill iteratively reason and reflect to ensure the generated formula is aligned more closely with the relation behind variables (lines 3-12), based on $C_m$ and the following components. The Domain-Sketch Generator guides LLMs to generate Domain-Sketch $S$, including a series of logical steps and descriptions (line 4). Next, the Code Generator turns $S$ into executable Python code $P$, which will be executed and generate imputed data $C'_m$ (line 5). If $C'_m$ is identical to $C$ based on the result of the Evaluator, $S$ and $P$ are considered correct and will be summarized into an imputation function $F$ for review and imported by the Executor (lines 6-7). Otherwise, the Reflector will refine the wrong sketch $S$ and generate a new Domain-Sketch $S_{new}$ for the Code Generator (lines 8-9). This process continues until the imputation Domain-Sketch is considered correct or reaches the *retry_limit* of Reflector. Note that, the *retry_limit* is a hyperparameter, where in our experiments *retry_limit=3*. If the retry times reach the *retry_limit*, the program will send feedback of unable to impute. It then executes the derived formula on the dirty data (line 13) and outputs the repaired data $D'$, as well as the function $F$ (line 14).

As demonstrated in Figure 3, SketchFill mainly includes four modules to carry out the DMVI task: Domain-Sketch Generator 3.3, Self-Reflected Code Generator 3.4, Summarizer 3.5, and Executor 3.6. We will discuss the details of each module in the following sections.

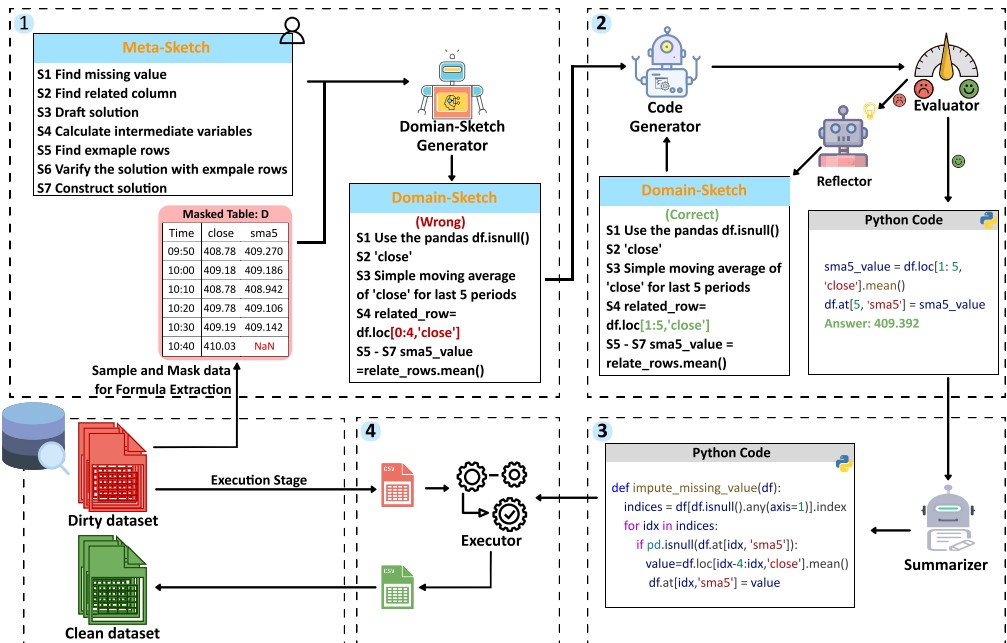

Figure 3: The SketchFill framework

## 3.3 DOMAIN-SKETCH GENERATOR

LLMs have demonstrated their ability to capture the two-dimensional structure of tabular data through techniques such as role-prompting and fine-tuning (Sui et al., 2024; Zha et al., 2023). However, DMVI requires a series of operations, such as locating missing values, building solutions and calculations. Even with step-by-step prompting of CoT reasoning, LLMs still struggle to fully comprehend the formulaic relationship and conduct calculations. To address this challenge, SketchFill includes the Domain-Sketch Generator to construct the imputation solution as one of main contributions that improve DMVI performance (See Phase ①, Figure 3).

The Domain-Sketch Generator is tasked with producing the Domain-Sketch, adhering to Meta-Sketch guidance to exploit the LLMs' embedding knowledge about the specific dataset at hand. The Meta-Sketch, a pseudo-code-like file, contains a series of logical steps and descriptions, encapsulating the formula-based strategy for DMVI. Then, LLMs could generate specific step-by-step Domain-Sketch to guide other agents to output Python code for DMVI. Consider the circumstance that domain experts of dirty data when imputing the missing value. It is common for them to decompose DMVI into the following steps, w.r.t the Meta-Sketch template in Phase ①, Figure 3:

- S1-S2: locating the missing value and recalling the associated knowledge about the variable of missing value, by searching for variables in this dirty table that are related to the missing value,
- S3-S6: drafting and verifying the formula to calculate the missing value by applying it to rows without missing values, and,
- S7: calculating the missing value, utilizing the verified formula.

In our experiments, we sufficiently illustrate that LLM-powered (See Appendix B.1) Domain-Sketch Generator can carry out a satisfying Domain-Sketch for the specific dataset based on the instruction from Meta-Sketch. To exemplify, given a masked clean subset ($C_m$), the Domain-Sketch Generator receives this $C_m$ as input and creates the Domain-Sketch ($S$) based on the Meta-Sketch.

## 3.4 SELF-REFLECTED CODE GENERATOR

The Code Generator is initialized as an LLM (see Appendix B.2) and serves as the interpreter of the Domain-Sketch ($S$) to generate Python code ($P$) for DMVI (See Phase ②, Figure 3). However,

writing code in a single attempt can be challenging, making it difficult to ensure the correctness of the imputation. Recent research has demonstrated that self-reflection can greatly improve answer accuracy, consistency, and entailment of LLM (Ji et al., 2023). Inspired by Reflexion and MetaGPT (Shinn et al., 2023; Hong et al., 2023), SketchFill incorporates a self-reflection mechanism that iteratively refines the initial Domain-Sketch ($S$) and generated code, enhancing its accuracy and reliability. Moreover, it reduces the need for human intervention in reviewing the DMVI solution. The Self-Reflected structure comprises two components: the Evaluator and the Reflector.

**Evaluator.** Within the Phase ②, the Evaluator holds a vital position by replacing the need for human evaluation during the imputation. The Evaluator receives the $C'_m$ as input and compares with $C$. Then, it returns a binary signal of whether the imputation is successful or not based on the result of *CloseMatch* defined as: $sgn(|C'_{m_{ij}} - C_{ij}| - \epsilon)$, where $\epsilon$ is a user defined hyperparameter that controls the tolerance of the closeness. If the imputation is regarded as identical, $P$ will be submitted to Summarizer for further processing. Otherwise, the Reflector shall be activated to refine $S$.

**Reflector.** The Reflector, initialized as an LLM (see Appendix B.3), is responsible for the self-reflected structure in SketchFill. To illustrate the mechanism behind, let us consider a scenario when the Python code ($P$) is generated due to a wrong $S$. The Evaluator then raise a negative signal and forwards the $S$ to the Reflector. The Reflector takes the $S$ generated in this iteration, identifying the root causes of the inaccuracies, such as incorrect index in Python code or wrong formula assumption (see Appendix C). Based on the analysis, the Reflector refines $S$ and generates a new $S_{new}$, as shown in Phase ②, Figure 3. This process repeats until it reaches the $retry\_limit$ or the Evaluator sends a positive signal, confirming the validity of $P$ for DMVI.

### 3.5 SUMMARIZER

The motivation for implementing the Summarizer stems from the necessity to conduct the human review for each imputation of subsets when applying LLMs for DMVI, owing to the token limitation that LLM can process at a time, particularly when utilizing CoT-based LLM. Meanwhile, LLMs have exhibited better factual consistency and fewer instances of extrinsic hallucinations in generating summary, compared to humans (Zhang et al., 2023b; Wu et al., 2023; Pu et al., 2023), as well as promising code understanding (Nam et al., 2023; Richards & Wessel, 2024).

The Summarizer is initialized as an LLM (see Appendix B.4) as part of SketchFill. Given the similar pattern of $P$ in the DMVI solution within the same column, the Summarizer has the potential to develop a generalized solution accordingly such that the Summarizer distil the $P$ into a more readable and generalized Python function. By doing so, the Summarizer drastically minimizes the need for human review. As depicted in Phase ③, Figure 3, the Summarizer takes the validated $P$ and abstracts the specifics of the dataset into parameters, creating a flexible function capable of addressing missing values across various subsets derived from the dirty table. Consequently, the Summarizer outputs a finalized Python function ($F$). $F$ is then utilized in the Execution module, enabling automatic DMVI across any subset of the original dirty table without additional manual procedures. The Summarizer thus streamlines the imputation process, ensuring consistency and scalability while significantly reducing the manual workload.

### 3.6 EXECUTOR

The Executor (See Phase ④, Figure 3) is programmed as a particularly Python file, aiming to process the dirty subsets with the missing value of the same columns by importing $F$ that is generated by the Summarizer. Therefore, once the $F$ is approved and deployed, the rest of the missing values can be automatically imputed. For instance, for the variable ($V$) in the tabular data case, the Executor is programmed to apply the DMVI processing with $F$ to impute subsequent missing value in $V$.

## 4 EXPERIMENT

### 4.1 ENVIRONMENT SETUP

**Approaches.** We compared SketchFill with the following approaches.

Table 1: Statistics of experiment datasets

|  | Bajaj | Bmi | Supermarket | GreenTrip | LOLChampion |
|---|---|---|---|---|---|
| #-Attributes | 12 | 5 | 10 | 13 | 12 |
| #-Variables | 6 | 3 | 6 | 4 | 5 |
| #-Tuples | 3600 | 720 | 960 | 1800 | 554 |
| #-Missing values | 334 | 138 | 180 | 215 | 107 |
| Missing values (%) | 9.28% | 19.17% | 18.75% | 11.94% | 19.31% |

(1) *KNN:* It is employed as an ML-based approach, where *N=5*. (2) *MICE:* It is employed as a statistic-based method, conducting DMVI by building chained equations of other variables. (3) *TabCSDI:* It is employed as a deep learning-based method, by utilizing diffusion models for DMVI. (4) *LLM:* It utilizes LLM intuitively. (5) *CoT:* It prompts LLM to reply with "Let's think step by step". (6) *Code Generation:* It requires LLM to generate complete code for DMVI. (7) *MetaGPT:* It leverages its built-in code generator and self-reflection module. The implementation details of non-LLM based approaches and MetaGPT are illustrated in Appendix D.

**Backend LLM.** We employ GPT-4o as the back-end model supported by OpenAI API[2]. As for open-source model experiments, we consider using the Llama3 (Llama3-8B-Instruct) model as the back-end[3]. The Llama3 request is supported by the Ollama[4] running in a local environment. For both models, the token size is set as 4096 using a temperature of 0.

**Dataset.** Our experiments span across five-domain datasets sourced from Kaggle and other open-access repositories. We illustrate the fact of the dataset and its formulas in Appendix A.

**Preprocessing.** Each dataset is sequentially segmented into subsets to accommodate the DMVI tasks that require proper calculation. Given the transparent formulaic association of the target variable to be imputed and other known variables, each subset is well curated as a testing tuple. Specifically, we mask the original value of the target variable with `NaN` to mimic the missing value controlled by an appropriate missing rate, similar to settings in other works (Zheng & Charoenphakdee, 2022; Tashiro et al., 2021; Van Buuren & Groothuis-Oudshoorn, 2011). Consequently, multiple missing values related to the same variable may take place in some testing tuples. Full statistics are shown in Table 1.

**Evaluation Metrics.** We assessed the imputation performance using three commonly adopted metrics: (1) *Accuracy:* This metric evaluates the overall accuracy by comparing the imputed values to the ground truth values. (2) *FindAccuracy:* This metric measures the accuracy of values that have been correctly imputed at least once, ignoring variables where all imputations failed. It is particularly helpful to evaluate datasets with noisy data, where a formula may not robustly hold for all rows but correct for the majority. It describes the accuracy of variable detection under the circumstance that the formula is constructed, even though not entirely correct. (3) *RMSE (Root Mean Square Error):* This metric calculates the difference between the imputed value and ground truth value. A lower RMSE indicates better imputation accuracy, with a zero value indicating identical imputation.

## 4.2 RESULT ANALYSIS

Figure 4 and Table 2 report the performance of DMVI using different approaches on five datasets from different domains. KNN and MICE approaches are statistic/ML-based; TabCSDI is deep learning-based; while the others are based on LLMs. We omit the imputation accuracy of KNN because its performance is less competitive. Besides, we only report the summary RMSE results of TabCSDI due to the limitation of its source code. Detailed results from Llama3-8B are shown in Appendix C. Next, we will analyze the experimental results for each dataset.

---

[2]https://platform.openai.com/docs/models/gpt-4o

[3]https://llama.meta.com/llama3/

[4]https://ollama.com/library/llama3

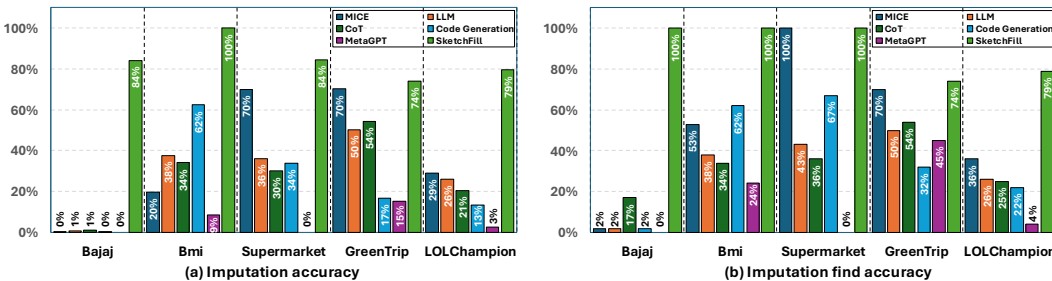

Figure 4: (a) SketchFill performance across 5 datasets showing imputation accuracy compared with different approaches. (b) SketchFill performance across 5 datasets showing imputation find accuracy compared with different approaches. LLMs are backend by GPT-4o.

Table 2: RMSE-measured imputation result by dataset by variable using different approaches

| Dataset | Variable | KNN | MICE | TabCSDI | LLM | CoT | Code Generation | MetaGPT | SketchFill |
|---|---|---|---|---|---|---|---|---|---|
| Bajaj | SMA5 | 1.1383 | 0.4232 | N/A | 0.5101 | 0.5051 | 0.7337 | 1.0743 | **0.0000** |
| | EMA5 | 1.6213 | 0.2014 | N/A | 0.3488 | 0.4004 | 0.8762 | 1.3908 | **0.0000** |
| | CCI5 | 69.5075 | 56.6780 | N/A | 59.7540 | 61.8278 | 76.1961 | 76.3084 | **0.0000** |
| | ROC5 | 0.5257 | 0.3239 | N/A | 0.3887 | 0.3943 | 0.5555 | 0.5779 | **0.0000** |
| | MOM10 | 3.1901 | 1.7620 | N/A | 1.8774 | 2.0001 | 2.6661 | 2.6459 | **0.0000** |
| | RSI8 | 7.8397 | **2.5590** | N/A | 4.9973 | 5.5522 | 7.1357 | 7.3535 | 8.7993 |
| | Summary | 13.9704 | 10.3246 | 17.2945 | 11.3127 | 11.7800 | 14.6939 | 14.8918 | **1.4666** |
| Bmi | Weight | 10.6149 | 2.7142 | N/A | 1.2109 | 0.8870 | 4.0040 | 14.9811 | **0.0000** |
| | Height | 0.0570 | 0.0424 | N/A | 0.0554 | 0.0671 | 28.6804 | 0.0649 | **0.0000** |
| | BMI | 4.6873 | 1.4047 | N/A | 0.0307 | 0.0342 | **0.0000** | 9.9793 | **0.0000** |
| | Summary | 5.1197 | 1.3871 | 5.0527 | 0.4323 | 0.3294 | 10.8948 | 8.3418 | **0.0000** |
| Supermarket | UnitPrice | 20.1411 | 18.5245 | N/A | 2.0401 | 6.1623 | 26.6527 | 26.6527 | **0.0000** |
| | Quantity | 2.2487 | 1.5734 | N/A | 0.0941 | 0.1374 | 2.2191 | 2.7918 | **0.0000** |
| | Tax5 | 9.3763 | **0.0000** | N/A | 0.8740 | 0.8962 | 2.9530 | 12.3807 | **0.0000** |
| | Total | 166.1456 | **0.0000** | N/A | 55.5336 | 38.8426 | **0.0000** | 246.1067 | **0.0000** |
| | CostsofGoodsSold | 146.9721 | **0.0000** | N/A | 0.5560 | 26.4411 | 239.0153 | 239.0153 | **0.0000** |
| | GrossIncome | 5.6514 | **0.0000** | N/A | 1.4392 | 0.6683 | 8.1654 | 8.9878 | 0.7749 |
| | Summary | 58.4225 | 3.3497 | 53.9156 | 10.0895 | 12.1913 | 46.5009 | 89.3225 | **0.1292** |
| GreenTrip | TipAmount | 2.6736 | 1.335 | N/A | 3.1168 | 3.3141 | 3.2938 | 3.2938 | **0.7329** |
| | TotalAmount | 3.4047 | 0.7834 | N/A | 1.5971 | 1.6945 | 12.6227 | 3.4435 | **0.5507** |
| | CongestionSurcharge | 1.1505 | **0.2636** | N/A | 1.2596 | 1.2596 | 12.0969 | 1.0591 | 0.3647 |
| | TollsAmount | 0.5387 | 0.7574 | N/A | 0.5540 | **0.4382** | 1.6866 | 0.5177 | 0.5720 |
| | Summary | 1.9419 | 0.7845 | 5.8039 | 1.6319 | 1.6766 | 7.4250 | 2.0785 | **0.5551** |
| LOLChampion | PRateplusBRate | 0.2008 | 0.0733 | N/A | 0.0474 | 0.0474 | **0.0466** | 0.2227 | 0.0474 |
| | D | 26.8432 | 14.6811 | N/A | 48.2048 | 43.4857 | 52.2125 | 46.1367 | **0.5847** |
| | K | 48.5434 | 45.2328 | N/A | 73.5042 | 180.9566 | 52.4258 | 71.2392 | **1.2712** |
| | KDA | 1.9720 | 1.8090 | N/A | 1.5361 | 2.1047 | 1.7919 | 1.7919 | **0.0267** |
| | PRate | 0.0670 | 0.0733 | N/A | 0.0730 | 0.0744 | 0.0716 | 0.0775 | **0.0329** |
| | Summary | 15.5253 | 12.3606 | 27.1722 | 24.6731 | 45.3338 | 21.3097 | 23.8936 | **0.3926** |

**Bajaj:** For this dataset, SketchFill exhibits exceptional performance in imputing variables governed by single-step formulaic relations such as SMA5, EMA5, ROC5, and MOM10 and demonstrates better accuracy in these variables with zero score of RMSE, although other approaches can still preserve a fair good imputation result at a low score. In the context of multi-step calculations required for variables such as RSI8 and CCI5, which present significant challenges for all approaches, it is noteworthy that SketchFill succeeded in imputing variables, such as CCI5, SMA5, EMA5, ROC5, and MOM10 with a zero RMSE score, surpassing other solutions. However, MICE approach performs best in terms of RMSE score on RSI8, as SketchFill fails to reason the correct formula of RSI8.

**Bmi:** SketchFill has effectively derived the BMI calculation formula for both forward and backward computations, achieving a 100% accuracy on all variables and zero scores on RMSE as shown in Figure 4. This flawless performance indicates that SketchFill can impute with remarkable accuracy when working on ordinary formulas. Besides, the Code Generation approach achieves all correct on the DMVI task of variable BMI with a zero score on RMSE.

**Supermarket:** We observe that SketchFill performs outstanding results on all variables, except for GrossIncome with an RMSE score of 0.7749, surpassed by MICE, which yields an RMSE score of 0. Notably, MICE achieves excellent results in terms of imputation find accuracy as demonstrated in

Figure 4. Also, contributed by the chained equations strategy of MICE, the imputation results on other variables, such as Tax5, Total, CostsofGoodsSold and GrossIncome, are all correct. Besides, the intuitive Code Generation approach is all correct on the DMVI task of variable Total with a zero score on RMSE.

**GreenTrip:** The GreenTrip dataset is collected from actual taxi trip records, which poses a wild challenge in that the formula behind involves more variables compared to other datasets. RMSE score illustrates that SketchFill also outperforms other approaches, except for TollsAmount surpassed by the CoT approach, also for CongestionSurcharge slightly surpassed by the MICE approach. Overall, SketchFill achieves the highest score of 74% for both accuracy compared to other approaches as demonstrated in Figure 4.

**LOLChampion:** The dataset LOLChampion was found to contain noisy data, such as duplication and ambiguous column names, posing challenges in the imputation process, particularly in formula derivation. Therefore, the Accuracy decreased to 79% for SketchFill, but still holds at least 50% higher accuracy than other approaches. Specifically, the dataset exhibits identical PRateplusBRate values across different Champion positions, despite having distinct PRate and BRate values. Thus, SketchFill performance was adversely affected, measuring RMSE score at 0.0474 (PRatepluseBRate) and 0.0329 (PRate) respectively. Nevertheless, SketchFill demonstrates superior performance with the lowest RMSE score in all variables except for PRateplusBRate.

## 4.3 DISCUSSION ON OTHER APPROACHES

The imputation accuracy of KNN approaches is limited across all datasets due to its weakness in in-context learning on tabular data. MICE achieves good performance in variables that are the linear combination of other variables because this approach conducts MVI based on the regression model. TabCSDI is a deep learning-based approach that leverages diffusion models for DMVI. However, its generative approach struggles to learn the formulaic relationship of numeric variables in the dataset. MetaGPT has developed a series of data-cleaning strategies (see Appendix D) for handling dirty data. However, these pre-defined strategies lack context awareness, leading to their struggles with DMVI tasks across all datasets.

## 5 CONCLUSION

This paper evaluates the performance of DMVI across several approaches using five datasets from various domains. The findings highlight the exceptional performance of our proposed approach, SketchFill, particularly in the context of single-step formulas. Notably, SketchFill achieves 74% Accuracy and 100% FindAccuracy on 3 datasets, with fabricated data inside the datasets. Furthermore, SketchFill demonstrates an overall accuracy of 84.2% across these datasets, demonstrating its robustness. And the Accuracy of SketchFill is 56.2% higher than CoT-based approaches, 59% higher than Code Generation approaches and 78.8% higher than MetaGPT. This establishes a new standard for automated data cleaning and points a new direction for missing value imputation. However, SketchFill is not incompatible with rendering non-derived features such as observation, transaction, or trading data (refer to the *close* price of the *Bajaj Finance* data from Figure 1). Additionally, SketchFill encounters difficulties in handling multi-step calculations and noisy data when processing Bajaj and LOLChampion datasets respectively. Furthermore, we observe that the performance of SketchFill relies on the inference capabilities of the backend LLM. As discussed in Appendix C, the Llama-8B model conducts limited capabilities on complex DMVI tasks. These issues present opportunities for future research on DMVI tasks using LLMs.

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

# A    DATASET BREAKDOWN

## A.1    FINANCE: BAJAJ

Bajaj Finance Stock Price Data with Indicators, with license: *CCO: Public Domain* (Debashis, 2023). It is originally sourced from a listed financial company in India and released on Kaggle. The testing dataset captures a historical period of stock exchange information at a 10-minute interval. The imputation results of all variables are evaluated under the *CloseMatch*, where $\epsilon = 0.001$.

**SMA5:** the abbreviation of the simple moving average of 5 periods. Here is the formula:

$$SMA5 = \frac{1}{5} \sum_{i=0}^{4} Close_{t-i} \tag{2}$$

**EMA5:** the abbreviation of the Exponential Moving Average of 5 periods. Here is the formula:

$$EMA(N) = \frac{2}{N+1} \cdot Close_N + (1 - \frac{2}{N+1}) \cdot EMA(N-1) \tag{3}$$

where $N = 5$ and $EMA(1) = Close_1$.

**CCI5:** an index to evaluate the stock market, formulated as:

$$
\begin{aligned}
CCI(n) &= \frac{(TP - MA(TP, n))}{0.015 \cdot MD} \\
MD &= \sqrt{\frac{\sum_{i=t-n}^{t}(Close_i - TP_i)^2}{n}} \\
MA(TP, n) &= \frac{TP_1 + TP_2 + \cdots + TP_n}{n} \\
TP &= \frac{(Height + Low + Close)}{3}
\end{aligned}
\tag{4}
$$

where $n = 5$.

**ROC5:** an index to evaluate the stock market, formulated as:

$$
\begin{aligned}
ROC(n) &= \frac{AX}{BX} \\
AX &= Close_t - Close_{t-n} \\
BX &= Close_{t-n}
\end{aligned}
\tag{5}
$$

where $n = 5$.

**MOM10:** an index to evaluate the stock market, formulated as:

$$MOM(n) = Close_t - Close_{t-n} \tag{6}$$

where $n = 10$.

**RSI8:** an index to evaluate the stock market, formulated as:

$$
\begin{aligned}
RSI(n) &= 100 - \frac{100}{1 + RS(n)} \\
RS(n) &= \frac{\text{Average of n day's up closes}}{\text{Average of n day's down closes}}
\end{aligned}
\tag{7}
$$

where $n = 8$.

## A.2 Health: Bmi

Age, Weight, Height, BMI Analysis (Missonnier, 2023). The dataset is listed on Kaggle for public access, although the license is not specified. The dataset comprises 741 individual records that cover attributes, such as height, weight, and BMI. The imputation results of all variables are evaluated under the *CloseMatch*, where $\epsilon = 0.01$.

**BMI, Weight, Height:** the abbreviation of Body Mass Index. Here is the formula:

$$BMI = \frac{Weight}{Height^2} \tag{8}$$

## A.3 Retail: Supermarket

Sales of a Supermarket, with license: *Apache 2.0* (Bansal, 2023). The dataset is originally sourced from the 3-month historical sales transaction of a supermarket company in Myanmar and released on Kaggle. For each entry, it covers the necessary fields, such as the unit price, quantity, and sale tax. The imputation results of all variables are evaluated under the *CloseMatch*, where $\epsilon = 0.01$. **Total,**

**UnitPrice, Quantity, Tax5:** here is the formula:

$$Total = UnitPrice \cdot Quantity + Tax5 \tag{9}$$

**GrossIncome, CostsofGoodsSold:** here is the formula:

$$GrossIncome = CostsofGoodsSold \cdot GrossMarginPercentage \tag{10}$$

## A.4 Transportation: GreenTrip

Trip Record Data (Taxi & Commission, 2024). The license is not specified on its website, but users don't have to submit an access request, which is now available for immediate download. The dataset is collected from taxi trip records in New York City and is actively updated every month. For each entry, it covers the necessary fields, such as tip fee, total paid, and tolls. Specifically, we seize the Green Taxi trip records in January 2024 for testing. The imputation results of all variables are evaluated under the *CloseMatch*, where $\epsilon = 0.01$.

**TotalAmount, TipAmount, CongestionSurcharge, TollsAmount:** here is the formula:

$$\begin{aligned} TotalAmount = FareAmount + Extra + MtaTax + TipAmount \\ + CongestionSurcharge + TollsAmount + ImprovementSurcharge \end{aligned} \tag{11}$$

where the other fields are necessary fields provided in the dataset as well.

## A.5 Gaming: LOLChampion

LOL Champion Stats (Elixir, 2024). The dataset is downloaded from a game hub website, which is provided free of charge, and is intended for use by analysts, commentators, and fans. Specifically, we seize the LPL data from Spring 2023 to Spring 2024 as a testing base. For each entry, it introduces a bunch of statistics about one game character. The imputation results are evaluated under the *CloseMatch*, where $\epsilon = 1$ for K and D, $\epsilon = 0.1$ for KDA, $\epsilon = 0.01$ for PRate and PRateplusBRate correspondingly.

**KDA, K, D:** KDA is a metric to evaluate the performance of the player or champion on average per game, K means the number of opponents the champion kills on average per game and D means the number of times the champion was killed on average per game. here is the formula:

$$KDA = \frac{K + A}{D} \tag{12}$$

**PRateplusBRate, PRate:** PRate means the rate at which the champion is picked, and PRateplusBRate means the sum of the rate at which the champion is picked and the rate at which the champion is banned. Here is the formula, namely:

$$PRateplusBRate = PRate + BRate \tag{13}$$

## B SKETCHFILL PROMPT TEMPLATE

### B.1 DOMAIN-SKETCH GENERATOR

Assume that you are a data scientist. I offer you a table in CSV form with missing values denoted as NaN. The first row is the variables' names it contains, and the separator of this CSV format file is char ",". Suggest a solution to fill in each missing value, denoted by NaN. **You must sketch your solution into the following template for each missing value you found**.
Process all the steps and Give Python code solutions for each missing value. This is extremely important. Omitting any steps of any missing value is forbidden.

**Step 1 Finding Missing value:** find the location of the missing value and describe the missing value, outputting the entire row where the missing values are located in this step.
**Step 2 Finding related Columns:** Find related Columns that are related to the missing value column you are filling. These related Columns are helpful for the imputation of missing values. Outputting the names of these related Columns in this step.
**Step 3 Drafting Solution:** Using the related Columns you find in Step 2, draft the solution for missing value imputation. The solution should be based on the related columns you find. Outputting the solution.
**Step 4 Calculating Intermediate Values:** Check if there were unknown variables in the solution. If there were, calculate the intermediate values of the intermediate Variable missing and needed in the solution. Output the calculation process of all the intermediate values in this step.
**Step 5 Finding Related Rows:** Find the values of other rows in the table that are needed in the imputation. Outputting all the values you find in this step.
**Step 6 Calculating and Verifying the parameters:** Check if there were unknown or unsure parameters in the solution for missing value imputation. You need to calculate and verify these parameters based on rows without missing values. Find 3 rows as examples for you to calculate and verify the parameters. Output the parameters you get and the rows you used in this step.
**Step 7 Use results from step 1 to step 6 and rebuild the Solution in Python code and combine all the steps and Python code you generated in this new Python code.** When you rebuild the code, you must make sure the value for imputation is in the same row and column of the missing value. Remember the index in Python is 0-based, the first number starts with 0. Generate the rebuilt solution in Python code way. So be extremely careful with the row index when rebuilding your Python code. And write your code in this format:

**### Python**
Your Python code for rebuilding the solution
**### Python**

Process all the steps and Give Python code solutions for each missing value. This is extremely important. Omitting any steps of any missing value is forbidden. **Here is the data:**
**{data}**

### B.2 CODE GENERATOR

**Assume you are a code rewriter, you are given a Python code sketch for imputation task on the given data.** The new Python code you rewrite should take the given data for input and fill in the missing value of it. When you rewrite the code, you must slice the dataset and use the same row or column index in the given Python code sketch. Trust the Python code in the given sketch. You must turn this data as DataFrame of pandas in your Python code. The Python code needs to save the dataset in csv format after imputation in this path {save_path}. Here is the requirement:

Give only the Python code for your reply. Do not generate any other information. And write your code in this format:

**### Python**
Put only your rewritten Python code here.
**### Python**

Here is the Python code sketch for you to rewrite:
**{code}**

You must turn this data as DataFrame in your Python code.
Here is the data: **{data}**

### B.3 REFLECTOR

> **You are an advanced reasoning agent that can improve based on self-reflection.** You will be given a previous sketch trial in which you were required to generate a solution for missing value imputation for the given dirty table. You were unsuccessful in imputing missing values in the dirty table for some reason.
>
> Here are some hints for your reflection:
> **1. using the wrong solution**, try to use your domain knowledge in the field related to this data and fill in the missing value with the calculation based on other variables
> **2. using the wrong rows or columns when generating the solution**, please reflect the rows and columns you used for imputation. For example, you should use data from the second row to the fourth row, but you use data from the first row to the third row.
> **3. remember the index in Python is 0-based.**
>
> Here is the wrong sketch to reflect:
> **{wrong_sketch}**
>
> Here is the dirty data:
> **{dirty_data}**
>
> Requirement:
> In a few sentences, **Diagnose a possible reason for failure or phrasing discrepancy.** Take the hints as examples and **Give a new sketch** for the missing value imputation of this dirty table. The new reflected sketch must follow the same steps as the wrong sketch, this is extremely important. You MUST Return your answer in this Format:
>
> **### Diagnosis:**
> Write your diagnosis here
>
> **### New Sketch:**
> Write your new sketch here

### B.4 SUMMARIZER

> **Assume you are a code summarizer**, you are given a code focus on the imputation of missing value in a particular dataset. Please summarize this code into a function, so it can take any dirty dataset with the same structure. The input of the function is the dirty dataset, {clean_data_save_path}.
>
> When you are summarizing the code, pay attention to the following situation:
> **1. You need to find the missing values index** of the dirty data in the Python function.
> **2. There can be more than 1 missing value** in the given new dirty data, when you rewrite the given code, make sure it can impute multiple missing values in the given dataset.
> **3.** Remember the location of missing values in the new dirty data is not the same as the code provided. **Change the fixed index of the provided code into indexes capable of any location.**
>
> Here is the requirement:
> The name of the function is *impute_missing_value*. Give only the Python code for your reply. Do not generate any other information. Do not write any explanation. And write your code in this format:
>
> **### Python**
> Put only your rewritten Python code here.
> **### Python**
>
> **Here is the code need to be summarized:**
> **{code}**

## C    EXPERIMENT NOTE

### C.1    CALL OF REFLECTOR IN GPT-4O EXPERIMENT ON SMA5 VARIABLE OF BAJAJ DATASET

```
# Last attempt, wrong
missing_sma5_value = data['close'][13:18].mean()

# New attempt, correct
missing_sma5_value = data['close'][14:19].mean()
```

**Diagnosis on Domain-Sketch:** The failure in the previous sketch occurred because the range of rows used to calculate the missing SMA5 value was incorrect. This is because the SMA5 for a given row is the average of the close prices for the last 5 periods, including the current one.

### C.2    CALL OF REFLECTOR IN LLAMA3-8B EXPERIMENT ON BMI VARIABLE OF BMI DATASET

```
# Last attempt, wrong
def calculate_bmi(weight, height):
    return (weight / (height ** 2)) * 703

# New attempt, correct
def calculate_bmi(weight, height):
    return (weight / (height ** 2))
```

**Diagnosis on Domain-Sketch:** The possible reason for failure or phrasing discrepancy is that the formula used to calculate BMI is incorrect and does not take into account the actual values in the Height and Weight columns.

### C.3    SKETCHFILL EXPERIMENT USING LLAMA3-8B

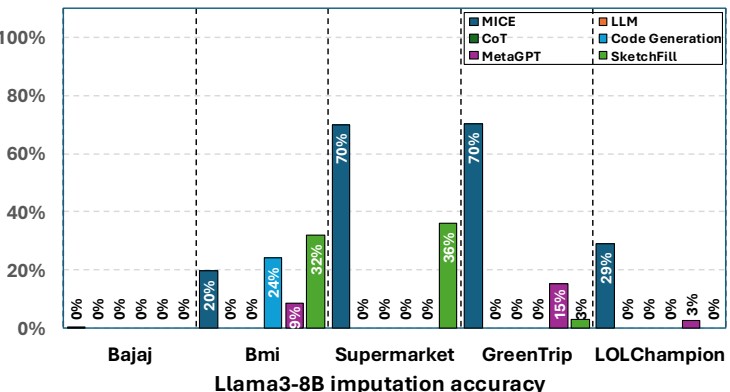

Figure 5: Llama3 imputation accuracy

We apply the same workflow settings as we configured on GPT-4o experiments. In light of the practical performance of llama3-8B, we slightly alter the prompt design to accommodate the model's capabilities, meanwhile retaining the identical framework in parallel. According to the imputation accuracy of MICE and MetaGPT approaches in Figure 4(a), SketchFill using Llama3 even achieves higher accuracy on the Bmi dataset. However, it is not as good as MICE on the Supermarket and GreenTrip datasets concerning the reasoning difficulties of the model itself on complex formula.

## D  ADDITIONAL IMPLEMENTATION

**KNN:** Thanks to the *scikit-learn* package that provides a KNN-based imputer, we utilize it to conduct the relevant experiment. Below is the code snippet:

```python
from sklearn.impute import KNNImputer
# import other packages ...

def knn_imputation(dirty_data_path, ...):
    dirty_data = pd.read_csv(dirty_data_path)
    missing_columns = ...
    imputer = KNNImputer(n_neighbors=5)
    dirty_data[missing_columns] = imputer.fit_transform(
        dirty_data[missing_columns])
```

**MICE:** We implement MICE method based on the original paper (Van Buuren & Groothuis-Oudshoorn, 2011). Although it is implemented in R, we find an alternative implementation in Python using *sklearn.impute.IterativeImputer*. More details can be found in the relevant documentation[5].

**TabCSDI:** We adopt its framework based on the original paper (Zheng & Charoenphakdee, 2022) and Github repository[6]. The diffusion model is trained and validated on our experimental datasets, and we ran our tests on internal server with NVIDIA 4090 GPUs.

**MetaGPT:** Thanks to the newly released toolkit *DataInterpreter* (Hong et al., 2024) in MetaGPT, we can easily deploy a LLM agent for the MVI testing. Below is a code snippet to demonstrate how we utilize it to perform MVI:

```python
import asyncio
from metagpt.roles.di.data_interpreter import DataInterpreter
# import other packages ...

async def meta(query):
    di = DataInterpreter()
    await di.run(query)

query = f"""Please read file from local file path: {dirty_data_path},
imputed the missing value,
save the imputed data file in path: {result_data_path}"""

asyncio.run(meta(query))
```

The experiment result has been combined into the Figure 4 and Table 2. Moreover, we go through the source code of the MetaGPT repository[7] and found that it completes missing values with simple strategies, such as *mean, median, most frequent*. Below is a code snippet to show how it works:

```python
class FillMissingValue(DataPreprocessTool):
    """
    Completing missing values with simple strategies.
    """

    def __init__(
        self, features: list, strategy: Literal["mean", "median",
        "most_frequent", "constant"] = "mean", fill_value=None
    ):
        self.features = features
        self.model = SimpleImputer(strategy=strategy, fill_value=
            fill_value)
```

---

[5]https://scikit-learn.org/stable/modules/impute.html
[6]https://github.com/pfnet-research/TabCSDI
[7]https://github.com/geekan/MetaGPT/blob/main/metagpt/tools/libs/data_preprocess.py

