# OpenReview forum: "SketchFill: Sketch-Guided Code Generation for Imputing Derived Missing Values"
_ICLR.cc/2025/Conference — ICLR 2025 Conference Withdrawn Submission_

### Official Review · Reviewer_EYmL · 2024-10-21

**Soundness:** 2
**Presentation:** 2
**Contribution:** 2
**Rating:** 3
**Confidence:** 5

**Summary:**

The paper develops a LLM-based code generator for missing value imputation. The key contribution is the use of a pre-defined sketch of a solution that the LLM iteratively refines and code generates for. Studies on small tables show that the approach does better than a knn-based approach and other LLMs.

**Strengths:**

- Good description of problem: Solving missing value imputation is critical.
- The proposed approach of generating code via systematic sketch input seemed to be well justified and the procedure automated human/data scientists was unique and novel.

**Weaknesses:**

* In particular, there is no study done to assess each of the components discussed in Fig 3. It would be good to understand how each of the components as they are built add to improved performance of the overall system.


* The lit. review seems to compare to some prior imputation approaches but evaluation is done withKNN as a  baseline. It would be good to compare to other statistical approaches like mean, conditional mean, Unconditional mean imputation, Conditional mean imputation (Buck's method), Stochastic regression imputation and  Maximum likelihood-based imputation. A table showing the characteristics of each techniques, their advantages and disadvantages would be useful to the end reader.

* The paper could be strengthened by sharing the python code generated for their different dataset. It is safe to assume that the python code is not generating the formulae for the different datasets especially BMI and Bajaj ?

* The evaluation is performed on datasets that are fairly small and not reflecting real world challenges. The largest dataset has total tuples of ~3.6K and columns of size 13.  What is the computational complexity of the approach ? There also several datasets available on the kaggle framework where these approaches could be benchmarked in a thorough manner.

* The presentation of results were confusing; It was not clear how accuracy was computed ? Was it 1-MAE ? Additional insights from the Table (results) were missing.
* The paper can be strengthened by  adding key details about the LLMs themselves in particular architectures, use of ICL, fine tuning and others ..
* The sketch based solution was template-driven for this solution; it was not clear how it would scale for any new large dataset with missing values.

**Questions:**

* Would the approaches scale to larger datasets beyond the ones studied?
* Were ablations performed with different LLMs, fine tuning, etc?
* Sharing an example of generated code and code changes after each phase would be helpful for the review.

---

### Official Review · Reviewer_MrmL · 2024-10-22

**Soundness:** 2
**Presentation:** 1
**Contribution:** 2
**Rating:** 3
**Confidence:** 3

**Summary:**

The paper introduces SketchFill, a framework that uses LLMs to impute derived missing values in tabular data through sketch-guided code generation. The method employs a two-level sketch system --- Meta-Sketch for capturing domain expertise and Domain-Sketch for dataset-specific instructions --- along with reflection and summarization components to generate and verify imputation formulas. Experiments demonstrate that SketchFill achieves higher accuracy than existing methods for formula-based missing value imputation (MVI).

**Strengths:**

1. SketchFill introduces a two-level sketch framework for Derived MVI (DMVI). The Meta-Sketch encodes general imputation strategies while the Domain-Sketch translates these into dataset-specific instructions, enabling LLMs to discover and implement mathematical formulas through code generation.

2. The method incorporates automated verification and refinement through an Evaluator-Reflector system. When imputation errors are detected, the system analyzes and revises the Domain-Sketch, creating a self-improving loop that requires no human intervention.

3. The paper addresses DMVI - a distinct challenge from standard imputation that requires discovering and applying formulas across multiple rows and columns. This task extends beyond statistical patterns to require understanding relationships between variables.

**Weaknesses:**

1. The related work section can be improved by expanding the discussion of LLM-based methods in greater detail. While the current section discusses CoT prompting, code generation, and MetaGPT, it lacks an analysis of how these approaches address the specific challenges of DMVI. For instance, discussing scenarios where CoT prompting outperforms code generation, or vice versa, in terms of handling imputation tasks would offer more clarity on the relative strengths and weaknesses of each method. This could also include the types of missing values and data structures each method is most suited for.
The section would also benefit from addressing any challenges that LLM-based methods still face in this domain, such as token length restrictions or susceptibility to producing hallucinated outputs in complex imputation tasks.

2. The novelty of the proposed method is somewhat outlined in sections 1-2, but the differentiation from existing methods could be made clearer. While the paper introduces SketchFill as a sketch-guided approach for DMVI using a combination of Meta-Sketch and Domain-Sketch for generating imputation formulas, the uniqueness of this approach could be made clearer by contrasting it more explicitly with the limitations of existing LLM-based methods.
For example, the authors could highlight that SketchFill offers a structured way to guide LLMs through multi-step reasoning processes by explicitly incorporating domain-specific knowledge, whereas existing methods struggle with tasks requiring complex derivation formulas or reasoning across multiple data points.

3. The paper lacks analysis of the Meta-Sketch's impact on performance - it's unclear how different Meta-Sketch designs affect SketchFill's ability to discover correct formulas. An ablation study comparing different Meta-Sketch structures, or experiments showing how variations in Meta-Sketch detail influence imputation accuracy, would help to understand which sketch components are needed.

4. The paper lacks analysis of SketchFill's performance in scenarios where a dataset contains both derived and non-derived missing values. The current evaluation focuses exclusively on formula-based missing values. The authors could extend their experiments to show how SketchFill compares with traditional imputation methods for non-derived values, or demonstrate how their Meta-Sketch system could be adapted to detect and handle different types of missing values within the same dataset.

**Questions:**

- Pg.1: Derived Missing Value Imputation (DMVI) is introduced as a term but not defined.
-  Fig. 1 is presented as an example of the derivation process, but the example is not explained within the text
- Fig. 2: How the Meta-Sketch and Domain-Sketch guide LLMs in generating more accurate code, or the purpose of the Reflector and Evaluator, is not explained well in Fig. 2 or the accompanying text.
- Fig. 3: The authors can improve the clarity of the figure by explicitly labeling the role of the Reflector in correcting errors within the Domain-Sketch and clarifying the sequential flow between generating Python code and applying it in the execution stage.
- Pg. 8: What is the univariate conditional model used in the MICE algorithm?

**Grammar**
- Pg. 1: “Missing value is a critical issue in data science”

---

### Official Review · Reviewer_2GL8 · 2024-11-01

**Soundness:** 1
**Presentation:** 2
**Contribution:** 1
**Rating:** 3
**Confidence:** 4

**Summary:**

This paper proposes a novel LLM-based method for missing data imputation, called SketchFill. The method emulates expert strategies through a pipeline termed Meta-Sketch, designed to effectively handle this task. At specific stages, the method utilizes the LLM to generate code approximating the imputation function and employs techniques, referred to as Reflector, to optimize this function further.

**Strengths:**

The writing in this paper is clear, and it includes several experimental results.

**Weaknesses:**

- In my view, the proposed method in this paper is overly focused on engineering aspects, and I struggle to identify a clear **scientific contribution**.
- The problem addressed in this paper does not align with the formulation of missing data. The authors focus on predicting values in a single target column using other columns as predictors, which essentially frames this as a very very very standard supervised learning task rather than a missing data problem. Calling this as a missing data issue solely because some values are absent can be misleading.
- The framework proposed in this paper essentially mirrors the standard procedures of supervised learning. Several steps in the Meta-Sketch, like S1 and S2, are very standard preprossing procedures, while in S3, common regressors such as linear models or neural networks are typically employed to predict the target variable, just as done here. S4 involves predicting outcomes, which aligns directly with conventional practices, and finally, validation is conducted in steps S5 and S6 to assess model performance. Overall, this approach closely resembles a standard supervised learning task.
- The core of this technique lies in **code generation**, which is used to approximate an unknown function. Typically, optimizers are applied to refine the model; however, this paper relies entirely on the LLM, including the "reflector" component, to optimize this function. There is no guarantee that this approach will yield an improved solution, as relying on the LLM as a definitive guide for optimization is werid and raises concerns.
- The effectiveness of the proposed method comes from its reliance on well-known criteria, such as BMI. For instance, when prompted to calculate BMI, the LLM can readily provide the equation based on the input variables. This effectively reduces the problem to a retrieval task. However, this raises the question of how the method would perform when tasked with approximating a more complex function, such as \( y = 238 \cdot x^2 \cdot 29 \cdot z + \exp(k) \cdot 22.33455454 \). Would this approach still be viable in such cases?
- Perhaps conferences focused on LLM, like ACL, would be better place for this paper, as they are likely to have reviewers better equipped to evaluate its contributions to LLM applications. However, from the perspective of a ML researcher, I still struggle to identify any **scientific contribution** in this work.

**Questions:**

see weaknesses.

---

### Official Review · Reviewer_wAZk · 2024-11-03

**Soundness:** 2
**Presentation:** 2
**Contribution:** 1
**Rating:** 3
**Confidence:** 4

**Summary:**

This paper proposes a machine learning model to impute missing values in tabular data. The authors' approach includes leveraging meta-information about column dependencies to train the LLM, integrating various generation methods, and imputing missing values through Python code generation.

The proposed method offers significant advantages, such as ease of implementation and flexible application. Notably, it enables the imputation of missing values by utilizing contextual information learned about table columns through LLMs, allowing it to learn structural constraints and fill missing values accordingly. This approach addresses one of the critical challenges in tabular data generation.

**Strengths:**

The paper provides a systematically engineered and comprehensive process for addressing the problem of imputing missing values in tabular data. The methodology, which leverages the pretrained knowledge within LLMs to learn dependencies among variables, is particularly intriguing. This approach shows a notable strength in table generation when hidden constraints are present, and its effectiveness is demonstrated through experimental data.

**Weaknesses:**

The proposed methodology does not fundamentally address or propose a new principle for table data generation, nor does it solve its intrinsic challenges, which means the learning of distributions from data. Instead, the approach is primarily based on Conditional Mean Imputation using masking techniques, with a strategy that validates the effectiveness of this method before extending it to other approaches. The proposed method is closer to an engineering ensemble approach of previously developed data synthesis methods.


Therefore, the observed superior performance likely results from the method’s ability to effectively impute missing values when there are physical constraints among column data. However, this situation is likely to be limited in practical tabular data applications, as missing values in data with such constraints are typically expected to arise only in specific cases, such as code errors, in most databases.

**Questions:**

1. In the proposed method, it is unclear how the system would handle such cases if the necessary columns for imputing missing values are absent. Conducting an ablation study on the results in Table 2 would likely provide valuable insights into the effectiveness and limitations of this approach. It may be helpful to construct a Bayesian network and compare the test error of constrained columns with the test error of their parent nodes.


2. Could this approach also be useful for generating tabular data with unknown constraints? It may be feasible to experiment with simulated data containing arbitrary constraints to test the method's robustness in such contexts. It would be beneficial to conduct reproducibility experiments using data with arbitrary nonlinear relationships in \( K \) or more dimensions. I suggest increasing \( K \) incrementally to verify the usefulness of the proposed method.

3. The experiments seem somewhat limited in demonstrating the utility of the proposed method. If incorrect relationships are provided via the LLM, can the model correct these through the data itself? Testing this aspect could clarify the method's reliability in scenarios where relationship information might not be entirely accurate. It would be helpful to modify the column information in the table data to see if the LLM misinterprets the column names, and then evaluate the performance of the synthetic data.

---

### Note · Authors · 2024-11-13

**Comment:**

We have decided to withdraw our submission from this round of consideration. We would like to extend our sincere thanks to you and each reviewer for the time, effort, and invaluable feedback provided. The constructive comments have given us a clearer perspective on areas for improvement, and we are grateful for the opportunity to learn from this experience.

**Withdrawal Confirmation:**

I have read and agree with the venue's withdrawal policy on behalf of myself and my co-authors.